## PERSPECTIVE

### C-tactile afferents: The mystery of human emotional touch has been hidden hair-deep

Ingvars Birznieks[1,2] 🟢

[1]*Physiology, School of Biomedical Sciences, Medicine & Health, UNSW Sydney, Sydney, NSW, Australia*
[2]*Neuroscience Research Australia (NeuRA), NSW2031 Randwick, Sydney, Australia*

Email: i.birznieks@unsw.edu.au

Handling Editors: Nathan Schoppa & Vaughan Macefield

The peer review history is available in the Supporting Information section of this article (https://doi.org/10.1113/JP289528#support-information-section).

Human skin is innervated by two major classes of sensory nerve fibres (afferents) – myelinated large-diameter A-fibres and small-diameter unmyelinated C-fibres. A-beta subclass afferents innervating innocuous (low-threshold) mechano-receptors in the skin are responsible for discriminative touch, allowing us to perceive various properties of objects, understand the environment and effectively control our movement. C-fibre afferents are typically mentioned in relation to nociception or pain perception. However, some C-fibre afferents are very different as they respond to very gentle, slow stroking of the skin. These so-called C-tactile (CT) afferents do not evoke specific conscious tactile perception. Instead, they represent a behind-the-scenes stealth emotional processing system (Schirmer et al., 2023).

The history of unveiling the existence of CT afferents and their physiological function in humans is intriguing and full of surprising discoveries. In animals, C-low-threshold-mechanoreceptors (C-LTMRs) associated with different types of hair follicles have been studied since early electrophysiological recording techniques from dissected nerves became available. It was assumed that with the significant transition from furry skin in ancestral animals to the relatively bare skin characteristic of modern humans this type of afferent might have experienced an evolutionary fading in favour of the fast-conducting A-beta tactile afferents.

In human research, the development of microneurography marked a significant breakthrough, enabling individual nerve fibre activity in healthy volunteers to be recorded by inserting the tungsten microelectrode through the skin and into the nerve. Single nerve fibre recording is achieved when the electrode tip is positioned and held close enough to isolate its electrical activity from other fibres.

The diameter of A-beta fibres is 6–12 µm while that of C-fibres is only 0.2–1.5 µm. As the microelectrode is manipulated by hand without any help of micromanipulators, the fact that such recordings are possible is truly astonishing. Obviously, it also explains why we became fully aware of CT afferents in humans some 20 years later after afferent types associated with A-beta fibres had already been well characterised. CT afferents responding to gentle touch in humans were first described in facial skin followed by systematic investigation in the hairy skin of the arm (Vallbo et al., 1993). We know now that in some cutaneous nerve branches in the forearm they may constitute up to 40% of all afferents encountered during recordings. Their responses appear to be tuned to a specific speed of gentle stroking which is about 3 cm/s, whereas rapid stroking reduces the response. Their exquisite tuning to slow, gentle stroking at skin temperature positions CT afferents as perfect mediators of hedonic (pleasant) touch, particularly in the contexts of social bonding and comfort (Schirmer et al., 2023). Indeed, evidence exists that CT afferent activation can facilitate oxytocin release, which is referred to as the bonding, trust, cuddle or even love hormone. Thus, CT afferents have been associated with 'affective touch' and pleasantness, but note that they do not directly convey any discriminative tactile sensation; instead, they can shape our emotional response to touch signalled by other types of touch receptors. However, their association with pleasantness might appear to be too simplistic as CT afferents are inferred to be involved in pain modulation and allodynia (a condition in which normally non-painful stimuli are perceived as painful) (Nagi et al., 2011). In fact, there might be several different types of CT afferents serving different function. In animals at least two different subtypes can be distinguished based on their molecular markers. Inter-estingly, one of these markers has been found in human scalp afferents, but so far, no molecular markers of CT afferents known in non-human counterparts have been identified in the rest of the body. Without molecular markers in humans, it has not been possible to trace these afferents and know where they terminate and what their receptor structure is.

The ground-breaking study by Moore et al. (2025) provides the first evidence demonstrating that in humans the CT afferents, just like in animals, can be directly associated with hair follicles and hair movement. It might be surprising that in over 35 years of research on CT afferents in humans their association with hair follicles had not been noticed. The fact that they have been found only in hairy skin was not seen as a hint, but rather as part of the puzzle.

Moore et al. (2025) demonstrate that responses of all 15 CT afferents they investigated were functionally linked to hair follicles and responded to slow manual deflection of the hair. Using optical coherence tomography the authors meticulously quantified skin surface displacement during hair deflection and concluded that perifollicular skin deformation caused by hair movement was too small and negligible to activate them and thus that stimulation must have occurred within the hair follicle itself. Moreover, mechanical hair removal (plucking) induced sustained firing which could exceed 10 s in duration.

The authors also observed a strong CT afferent response to spontaneous piloerection. It is tempting to speculate whether such reciprocal interactions with emotional goosebumps may explain the associated experience of psychogenic chills.

Does this mean we have solved the CT afferent enigma in humans? No, not really. Without molecular markers we do not know how many distinct CT afferent types there might be in humans. Are they all associated with hair follicles? How do we explain that on rare occasions they have been found in the glabrous (non-hairy) skin as well (Watkins et al., 2020)? Is pleasure mediated by a different type of CT afferent from those involved with shaping pain perception or is it determined by the context or specific discharge pattern? We should anticipate many intriguing discoveries yet to be made.

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

## Additional information

### Competing interests

No competing interests declared.

### Author contributions

Ingvars Birznieks: Conception or design of the work; Drafting the work or revising it critically for important intellectual content; Final approval of the version to be published; Agreement to be accountable for all aspects of the work

### Funding

Australian Research Council (ARC): Ingvars Birznieks, DP230100048; Australian Research Council (ARC): Ingvars Birznieks, DP200100630.

### Acknowledgements

Open access publishing facilitated by University of New South Wales, as part of the Wiley - University of New South Wales agreement via the Council of Australian University Librarians.

### Keywords

affective touch, C-LTMR, C-tactile afferents, hair follicle, mechanoreceptors

### Supporting information

Additional supporting information can be found online in the Supporting Information section at the end of the HTML view of the article. Supporting information files available:

**Peer Review History**

