## [Peer Review History · The Journal of Physiology]

Title: C-Tactile afferents: The mystery of human emotional touch has been hidden hair-deep

Ingvars Birznieks
DOI: 10.1113/JP289528

Corresponding author(s): Ingvars Birznieks (i.birznieks@unsw.edu.au)

The following individual(s) involved in review of this submission have agreed to reveal their identity: Andrew Marshall (Referee #1)

Review Timeline:

Submission Date: 01-Jul-2025

Accepted: 10-Jul-2025

Senior Editor: Nathan Schoppa

Reviewing Editor: Vaughan Macefield

Transaction Report:

Dear Professor Birznieks,

Re: JP-P-2025-289528 **"Title: C-Tactile afferents: The mystery of human emotional touch has been hidden hair-deep"**
by Ingvars Birznieks

We are pleased to tell you that your paper has been accepted for publication in The Journal of Physiology.

However, we have one formatting request: the reference list must be in alphabetical order, rather than numbered, to comply with our Journal format.

Please could you email a new file for publication by replying to this email? You could also correct the minor typo noted by the reviewer in the comments below.

The new manuscript provided will be used by the Production Editor to prepare your proof. When this is ready you will receive an email containing a link to Wiley's Online Proofing System. The proof should be thoroughly checked and corrected as promptly as possible.

Yours sincerely,

Nathan Schoppa
Senior Editor
The Journal of Physiology

If you would like to receive our 'Research Roundup', a monthly newsletter highlighting the cutting-edge research published in The Physiological Society's family of journals (The Journal of Physiology, Experimental Physiology, Physiological Reports, The Journal of Nutritional Physiology, and The Journal of Precision Medicine: Health and Disease), please click this link, fill in your name and email address and select 'Research Roundup':
<https://www.physoc.org/journals-and-media/membernews>

- You can help your research get the attention it deserves! Check out Wiley's free Promotion Guide for best-practice recommendations for promoting your work at: www.wileyauthors.com/eoo/guide. You can learn more about Wiley Editing Services which offers professional video, design, and writing services to create shareable video abstracts, infographics, conference posters, lay summaries, and research news stories for your research at: www.wileyauthors.com/eoo/promotion.

The Corresponding Author will receive an email from Wiley with details on how to register or log-in to Wiley Authors Services where you will be able to place an order

REFeree COMMENTS

Referee #1:

The article is accurate. There is a typo on line 57 which should say Moore and not Morre

EDITOR COMMENTS

Reviewing Editor:

Thank you for submitting your Perspectives article to The Journal of Physiology. I'm pleased to report that it is acceptable for publication, but I just need you to correct the typo on line 57 (Moore, not Morre)

Senior Editor:

Thank you for writing the Perspectives on the research manuscript by Moore and co-workers. It has been reviewed by one of the authors of that paper and we are happy to inform you that your working is considered to be acceptable for publication. As noted by the referee, there is one typo that needs to be corrected but this can be done in the Proofs stage.